# Advances in the Generation of Constructed Cardiac Tissue Derived from Induced Pluripotent Stem Cells for Disease Modeling and Therapeutic Discovery

**DOI:** 10.3390/cells13030250

**Published:** 2024-01-29

**Authors:** Truman J. Roland, Kunhua Song

**Affiliations:** 1Heart Institute, University of South Florida, Tampa, FL 33602, USA; trumanroland@usf.edu; 2Department of Internal Medicine, University of South Florida, Tampa, FL 33602, USA; 3Center for Regenerative Medicine, University of South Florida, Tampa, FL 33602, USA

**Keywords:** cardiac organoid (CO), engineered heart tissue (EHT), heart-on-a-chip (HoC), cardiovascular disease (CVD), induced pluripotent stem cell (iPSC), tissue engineering, organ modeling, disease modeling, drug discovery, regenerative medicine, self-assembly, high-throughput screening

## Abstract

The human heart lacks significant regenerative capacity; thus, the solution to heart failure (HF) remains organ donation, requiring surgery and immunosuppression. The demand for constructed cardiac tissues (CCTs) to model and treat disease continues to grow. Recent advances in induced pluripotent stem cell (iPSC) manipulation, CRISPR gene editing, and 3D tissue culture have enabled a boom in iPSC-derived CCTs (iPSC-CCTs) with diverse cell types and architecture. Compared with 2D-cultured cells, iPSC-CCTs better recapitulate heart biology, demonstrating the potential to advance organ modeling, drug discovery, and regenerative medicine, though iPSC-CCTs could benefit from better methods to faithfully mimic heart physiology and electrophysiology. Here, we summarize advances in iPSC-CCTs and future developments in the vascularization, immunization, and maturation of iPSC-CCTs for study and therapy.

## 1. Introduction

Cardiovascular disease (CVD) causes one-third of all deaths [1,2,3], leading global mortality [4,5,6,7,8,9,10,11,12,13,14,15,16,17], with myocardial infarction (MI) contributing the most among CVDs [6]. Heart failure (HF) is a terminal CVD resulting in irreversible loss of heart function with progression exacerbated by prolonged inflammation [6,15,18,19,20]. Compounding these are risk factors like diet, activity, environment [2,16,21], and infections like COVID-19, which increase the risk of developing a CVD or experiencing CVD-related complications [4,18,19,22]. Prolonged and/or excessive inflammation frequently occurs following environmental stress accumulation and CVD pathogenesis. Following MI, inflammation causes cellular dysregulation; harmful reactive oxygen species (ROS) accumulation [6,8]; the suppression of angiogenesis [6]; and ECM-degrading protease upregulation, largely matrix metalloproteinases (MMPs) [11,13], ultimately leading to adverse fibrotic remodeling [4,5,6,11,13].

Efforts to understand and treat CVD face the challenges of complex pathogenesis and the heart’s absence of regenerative capacity. Limited therapeutic options necessitate reliance on scarce donor organs and immunosuppression [8,10,11,12,16,23,24,25,26,27]. Preclinical studies have attempted to remuscularize infarcted hearts by regenerating new cardiomyocytes (CMs) from stem cells and non-CMs or by modulating the proliferation–maturation axis of existing CMs [1,17,26,27,28,29,30,31], the constituent cells of cardiac muscle [1,31] (Figure 1). Recent studies have also aimed to revascularize damaged tissue or reprogram fibrotic scar tissue into healthy musculature and vasculature [3,6,8,12,15,20,26,27,30,32,33]. Despite continued efforts, cellular therapies encounter viability issues. Infusions and injections of cell solutions containing mesenchymal stem cells (MSCs), iPSCs, and iPSC-derived CMs (iPSC-CMs) have shown modest recovery in some in vivo models [3,8,12] but highly variable results in human clinical trials for CVD, as poor cell retention and host integration challenge long-term treatment efficacy [1,3,8,10,12,27]. Over 99% of transplanted cells in solution quickly die for various reasons: host immune reactions, inflammation, protease release, low cell density, deficient cell–cell interactions needed to promote cell survival, and poor cell-type diversity, which limits host integration [1,10,27,30,34,35]. Critically, transplanting immature tissue poses an arrhythmia risk from improper CM electrical integration [10,24,27,30,31,34]. Even acellular factors face rapid clearance following treatment [10,30,34,35]. Capturing the complexity of pathological and wound-healing pathways in vitro is challenging [36]. Considerable effort has gone into making therapies more viable and targeted, yet much is still left to be desired [7].

Inducing pluripotency in human cells obtained through skin biopsy, blood draw, urine samples, or hair follicles [8,37], combined with CRISPR gene editing, has facilitated the rapid development of constructed cardiac tissues (CCTs) for disease modeling and therapeutic discovery [4,5,38,39,40,41]. Here, we refer to 3D in vitro cardiac tissue platforms as CCTs, including so-called cardiac microtissues (CMTs), cardiac organoids (COs), heart-on-a-chip (HoC), and other engineered heart/cardiac tissues (EHTs/ECTs) [5,8,23,31,34,38,42,43,44,45,46] (Figure 2). Induced pluripotent stem cells (iPSCs) can differentiate into virtually any cell type and simultaneously grow into multiple cell types (co-emergence) through controlled factor supplementation [7,24,37,39,46,47,48,49,50], environmental signals, and cell–cell interactions [39,43,46,51]. The differentiation of iPSCs into cardiac cell types is achieved by manipulating cardiogenic signaling, including the Wnt and BMP pathways [5,8,23,32,37,47,50,51,52,53,54,55,56]. CMs in iPSC-CCTs initially have a fetal phenotype. With proper environmental cues, CMs transition from a proliferative fetal phenotype into a mechanically robust, non-proliferative adult phenotype following endogenous YAP-Hippo pathway modulation [27,29,30,31,37].

Recapitulating native heart tissue in vitro is challenging because of the crosstalk between different cell types and the heart’s unique architecture [7,57]. The heart has three layers (internal to external: endocardium, myocardium, and epicardium) that are responsible for the continuous nourishment, contraction, and maintenance of cardiac structure (Figure 1B) [31,46,58]. CMs compose heart muscle and represent 70%/30% of fetal/adult heart cells, respectively, while occupying 70% of its volume [7,25,36,37,54]. Retinoic acid is a key factor for MYL7^+^ atrial CMs, whose action potentials (Aps) are much shorter than MYL2^+^ ventricular CMs [25,59,60]. The endocardium is composed of two major types of cells, endocardial cells (EndCs) positive for NFATC1 and endothelial cells (ECs) positive for PECAM1/CD31 [13,32,36,37,54,61] (Figure 1). ECs and EndCs compose ~40% of adult heart cells [62]. The epicardium forms via the fusion of the foregut-adjacent proepicardial organ with the developing heart in utero. Epicardial cells (EpiCs) expressing WT1 envelop the developing heart, also creating the pericardial cavity [7,52], composing 10% of adult heart cells [47]. EpiCs give rise to cardiac fibroblasts (CFs), vascular smooth muscle cells (VSMCs) (the most abundant cell type in vessels [13]), and pericytes (PCs), which migrate into the myocardium [32,45,63,64]. CFs, expressing TCF21 and sensitive to bFGF induction, maintain connective tissue and aid in the electrical integration of CMs by promoting the formation of the gap-junction protein Connexin 43 (Cx43) [25,36,37,41,54,64] and compose ~12% of adult heart cells [62]. MYH11^+^ VSMCs and NG2^+^ PCs, sensitive to PDGF-BB induction, compose the mural cell population, which supports vessels outside the EC membrane [10,13,15,20,25,32,36,37,64] and compose ~5% of adult heart cells [62] ECs release vasodilators like nitric oxide and vasoconstrictors like endothelin-1, which instruct VSMCs to relax or contract to modulate blood pressure [1,25,65].

Monoculture in 2D is quick and easy but dramatically restricts tissue crosstalk and fails to facilitate physiological cell–cell interactions [3,4,5,7,16,24,28,36,45,46,47,48,49,66]. Cells are interdependent, requiring sufficient contact density to promote cell survival and different cell types to establish homeostatic signaling feedback. Monolayer tissue culture fails to offer 3D cell–cell connections, reducing tissue complexity and physiological relevance. Even if several cell types from different lineages are co-cultured, they demonstrate lower tissue coordination and definition compared with co-emerged tissues [65], for instance, those arising from in vivo embryogenesis and the in vitro self-assembly of iPSC-composed embryoid bodies (EBs) [7,9,24,34,36,45,65,67]. A lack of structure can impair a model’s ability to detect drug-induced cardiotoxicity, a critical test for chronically and/or systemically administered drugs [4,8,16,34,36,41,45,46,54,67,68]. This was observed by Mills et al. during drug screenings in which successful results in 2D culture failed to translate to 3D culture [17,44]. Three-dimensional tissue culture further demonstrates its potential in markedly improved tissue maturation, a feature essential in modeling adult disease and producing transplantable tissue. iPSC-CCTs, like scaffolded EHTs [7,9,25,31,34,54,67] and scaffold-free/-seeded COs [16,30,36,39], possessing 3D architecture, far more effectively facilitate the transition of CMs into adult phenotypes [69] (Figure 2).

iPSCs are powerful tools for 3D human tissue models of healthy organogenesis and disease pathogenesis. Patient-specific iPSC derivation has substantially contributed to making personalized studies and drug discovery a reality [2,3,4,5,21,34,36,44,45,50]. Individualized drug screenings have the potential to address the variability in patient-to-patient treatment responses. In addition, iPSCs parallel the development of tissue from an embryonic state, giving rise to human congenital heart disease (CHD) models [4,7,34,36,51,55,67,70]. An increasingly popular method of iPSC-CCT self-assembly begins with an EB composed of centrifuge-pelleted iPSC aggregate [24,30,39]. EBs give rise to 3D tissues with abundant crosstalk between multiple cell types originating from the same lineage. This multi-directional growth frequently produces “miniature organs”, referred to as organoids or COs [43,47,51,56,71,72,73]. These CO platforms have been used to model developmental and adult-onset CVD [8,47,60]. iPSC bioengineering provides unprecedented opportunities to construct organogenic tissue models specific to patient genetics, age, and gender [5,34,36,39,40,46]. iPSC-CCTs have been applied to translational studies of cardiomyopathy, cardiac arrhythmias, and defects in cardiogenesis [3,4,5,14,30,34,37,39,41,45,46,47,53,56,59,60,67]. Self-assembled iPSC-CCTs can even produce 3D tissues on potentially batch-scales, offering life-like, translatable platforms for organ modeling, drug discovery, and regenerative medicine [1,10,17,18,34,43,44,56,65] (Figure 3).

## 2. Constructed Cardiac Tissues (CCTs)

### 2.1. Engineered Heart Tissues (EHTs)

EHTs are scaffolded CCTs designed to physically stimulate geometric cell-seeded tissue. The most basic are conventional scaffolds that offer substrate cues, while more advanced EHTs often suspend scaffolds between double elastomeric posts under mechanical tension, cyclic loading, and/or electrical stimulation [7,9,18,23,30,41,45,53,54,58,67,74]. Various geometries exist, including strips, rings, patches/films, tubes, and chambers [7,10,14,41,46]. EHT scaffold selection is critical. Hydrogel scaffolds using natural polymers are popular. Natural polymers (typically proteins or glycans) are very biocompatible but often lack durability [7]. The proteins most often used include collagen [3,4,29,36,45,46,66,75], gelatin (denatured collagen) [15,66,75], fibrinogen, fibrin (polymerized fibrinogen monomers) [3,4,11,36,46,66], mouse tumor-derived Matrigel [3,4,36,42,46,48,51,67,75], the adhesion proteins fibronectin and laminin [7,36,76], and even silk [77]. Glycans like hyaluronic acid (HA) [11,32,41,66,78], alginate [11,41,66,75], and chitosan [2,3,7,11,41] are popular, with agarose sometimes used [22,41,75]. Decellularized ECM (dECM) obtained from animal or human donor heart tissue is still frequently used [7,10,24,36,58,75,78]. dECM largely retains its native ECM structure following detergent decellularization and subsequent cell reseeding. This provides life-like matrix cues that promote electrical and metabolic maturation for iPSC-CMs in native heart dECM [7,24,25,36,41,58]. Microvessel matrices obtained from adipose lipoaspirate tissue can serve as pre-fabricated vascular templates in EHTs [41,65].

Synthetic polymers have fine-tunable properties but alone fail to facilitate cell adhesion [7]. Frequently used synthetic polymers may be categorized into polyesters, polyethers, polyolefins, and siloxanes. Polyesters are well established in biological, tissue engineering, and CCT applications. Thoroughly demonstrated biocompatibility and a degradation time of days to weeks make polyesters staples of synthetic scaffold hydrogels. Controlled payload release is achieved through the hydrolytic cleavage of ester groups. Commonly used polyesters include PLGA (and its copolymers, PLA and PGA) [3,10,36,41,75], PCL [10,15,36,41], PGS [15], and PoMaC [3,21,74]. Polyethers like PEG [3,45,58,66,75] and PU [7] and polyolefins such as PAm and PINIPAAm [58,75,76] are also utilized for hydrogel fabrication. Siloxanes stand out for their hydrophobicity and can function as surface substrates [26,74]. Polydimethylsiloxane (PDMS) is often used in micropatterning molds [57,58,75], biowire EHTs [74], EHT posts [2,8], and microfluidic HoCs [4,45,75,76], though there is interest in moving away from PDMS because of the absorption of hydrophobic drugs, with silicon, ceramics like glass, and thermoplastics like polymethylmethacrylate (PMMA) suggested as alternatives [45,75]. Hybrid scaffolds offer a more life-like ECM stand-in than single-polymer hydrogels, evidenced by improved tissue maturity [36]. Polymers can be mixed or chemically modified. Cell-adhesion and ECM-mimicking motifs like the RGD peptide, the RGDFK peptide, the YIGSR peptide, polydopamine (PDA), heparin, etc., are frequently attached to synthetic polymers [2,11,76]. PEG-RGD is a popular example [3,45,66]. Cross-linking modifications can improve stiffness and durability with self- or cross-reactive molecular species [7,10,36,58,66,75,78]. Methacrylate (MA) is a common cross-linking modification, often applied to HA and PEG [2,11,75] as gelatin-MA (GelMA) [2,11,14,41,75] and MA-HA [78].

Several scaffold fabrication methods have been developed in addition to the aqueous reconstitution method used in many simple hydrogels. Micropatterning can guide cell sheet growth for modular assembly or recapitulate native tissue patterning [4,43,58]. Soft-lithographic molds fabricated via stereolithographic (SLA) 3D printers are used to produce microscale patterns in PDMS. Micropatterns guide cell migration through surface grooves or adhesion-promoting biomaterial coatings [3,10,26,79]. Biomaterial-coated micropatterns can be stamped onto other surfaces to direct specific cell–cell interactions, including vascular patterning [24,25,58,79] (Figure 4). Such patterns can capture cardiac muscle anisotropy by directing CM elongation and sarcomere orientation [10,24,25,54,58,79]. Electrospinning is a versatile approach to customizable and modular scaffolds [25,28,80]. The application of high voltage to an extruding nozzle ejects thin polymer fibers, producing fibrous meshes or aligned fiber scaffolds. Fiber properties like thickness, alignment, and composition can be finetuned by varying extruder voltage and/or polymer formulation [3,10,25,66,80].

Bioprinting is an emerging EHT fabrication method that uses 3D printers to deposit “bioinks” composed of hydrogel, biomaterial, and cell mixtures into tissue-like structures. Multiple bioinks are frequently used per print, sometimes with multiple nozzles for co-extrusion. Notably, bioprinting is capable of producing macroscopic, life-size heart architecture [4,10,25,43,45,66,67,68,80], even printing fibrotic scar tissue [7]. Macroscopic models are important for characterizing heart function and output [14], and few methods currently match the anatomical scale achieved in 3D bioprinting using the freeform reversible embedding of suspended hydrogels (FRESH) technique [10,11,14,36,76]. Bioprinting has produced ventricles as externally large as in adults but with thinner, relatively fragile walls, incomplete CM coverage, and ~2% of the ejection fraction in adult human hearts [10,45]. With further development, bioprinting offers a promising avenue for developing macroscopic heart models. Sacrificial writing into functional tissue (SWIFT) is a bioprinting technique used to obtain vascular-like channels. Sacrificial bioinks are printed but dissolve in aqueous culture to enable fluid flow [7,10,14,24,36,45,61,75,76,80] (Figure 4).

EHTs excel in dynamic macroscale stimuli, providing multifaceted developmental cues difficult to implement in conventional culture [7,25,30,34,45,67]. Despite limited biological complexity, typically using cell seeding instead of self-assembly, EHTs offer exceptional CM maturation and are well suited to studying the mechanical and electrophysiological properties of cardiac tissue [7,9,25,31,34,54,67]. Cardiac stimulators [30,38,81] or internal electrodes may be used for electromechanical conditioning regimes. For instance, biowires can run along inside the EHT with chamber-specific phenotypes (atrial and ventricular) at opposite ends [7,34,74]. Electrical stimulus mimics the sinoatrial (SA) node’s master AP rhythm, and starting EHT stimulation earlier improves maturity [4,30,81]. Mills et al. demonstrated that the double-post-mechanical stimulation regimes of EHTs can be applied to more biologically complex platforms like COs/organoids for improved maturation and drug sensitivity [18]. EHTs reliably induce the mechanical maturation of CMs, extending cell and sarcomere length, forming T-tubule networks for excitation–contraction coupling, and improving calcium and AP handling [7,29,30,41,54,82] (Figure 5).

### 2.2. Cardiac Organoids (COs)

Spheroids are an important tissue engineering milestone, giving rise to robust platforms such as iPSC spheroids and cardiac spheroids (CSs). Unlike organoids, spheroids are generally simple aggregates of one or more cell lines, often terminally differentiated [9,25,26,27,28,29,30,31,32,33,34,35,36,37,38,39,40,41,42,43,44,45,46,47,48,49,50,51,52,53,54,55,56,57,61,75,76,83,84]. CSs may be added to existing organoids to introduce new cell types and structures, even mimicking the fusion of the proepicardial organ with the developing heart [56]. This “CCT building block” technique is used for co-culture with MSCs, macrophages (MFs), and heart layer tissue to produce more complete systems [7,36,43,56,61]. Spheroids have the advantage of well-defined ratios of cell populations [9,14]. While spheroids are commonly used as tumor models given their hypoxic core [48,75,85], this hypoxic gradient can be useful in modeling cardiac ischemia [4,14,16,46].

Organoids are 3D self-assembled tissues emerging from scaffold-free or scaffold-seeded cellular aggregates. Self-assembly occurs spontaneously through cell–cell and cell–matrix interactions. The recapitulation of organogenesis is frequently guided through simple factor supplementation [2,7,8,9,57,65]. Multilineage co-culture may be used given the speed of monolayer pre-differentiation [38,45,54,67], but co-emergence from EBs generally promotes better tissue crosstalk and organization [7,9,24,34,36,45,65,67], evident in improved vessel network stability from co-emerged versus co-cultured vascular organoids [65] (Figure 3). Self-assembly from EB has become increasingly popular [24,30,39], contributing to several remarkable achievements in CO development [43,47,51,56,71,72,73].

Several recent groundbreaking protocols have been published detailing the induction of pluripotent aggregates into cardiac mesoderm lineages whichself-assemble into COs. These COs produce several co-emergent cell types, chambers, and capillary-like structures. All three heart layer lineages have been established across and within individual reports as well [7,23,43,47,51,56,71,72,73]. Researchers have developed bi- and triphasic Wnt modulation protocols (up-, down-, and then optionally upregulated again) using the Wnt activator CHIR-99021 (CHIR) and the Wnt inhibitors IWP2, IWR, and Wnt-C59. Exogenous Activin A, BMP-4, and bFGF/FGF-2 were added during the initial Wnt upregulations for TGF-β superfamily activation and cardiac specification [56], later simplified to a minimum of ~1 ng/mL Activin A and BMP-4 [36,47,55], bFGF optional [42]. These protocols produce diverse cardiac cell populations of CMs, ECs, CFs, plus EndCs [23,51,56], EpiCs (often through the co-emergence of endodermal tissue) [23,43,52,63,67], or both EndCs and EpiCs [47]. Obtaining significant epicardial and VSMC/PC populations for vascular reinforcement produces particularly robust COs [16,20,45,47,71]. Microvascular and/or capillary networks (the initial EC tube networks that serve as the inner layer of mature, multi-layered vasculature, occasionally with non-contractile PC reinforcement) and better tissue differentiation could be achieved with VEGF-A supplementation [10,45,56,63,73,86] (Figure 4).

Ventricular- and chamber-forming COs have been developed via biphasic Wnt modulation [56,86]. All COs receive exogenous bFGF and BMP-4 to produce CMs and some ECs, though the further addition of VEGF-A produces larger EC and CF populations in distinct layers. EpiC CS aggregates are added and partially envelope the COs [56]. Harnessing crosstalk between embryonic germ layers, both Silva et al. and Branco et al. achieved epicardium and myocardium induction through mesodermal–endodermal co-emergence [52,71], while Drakhlis et al. used mesodermal–endodermal foregut co-emergence to establish the endocardium and myocardium [51,52], establishing the beginnings of an EC vascular network through para- and juxtracrine interactions with developing foregut tissue [51,71]. Lewis-Israeli et al. achieved multiple, interconnected chambers in COs possessing all three heart layers from a single mesodermal lineage using triphasic Wnt modulation, boasting several populations of important supporting cells like ECs, CFs, and EpiCs, together resulting in microvasculature [45,47]. While Lewis-Israeli et al. did not report staining for mural markers, Silva et al. also achieved significant epicardial populations and reported microvasculature stabilized by PCs and CF-like VSMCs. These studies suggest that substantial EpiC and epicardial-derived cell populations (e.g., CFs and mural cells like VSMCs/PCs) hold promise in developing mature vasculature through vessel stabilization and smooth muscle reinforcement [10,45,47,50,65,71] (Figure 4). CO technology continues to evolve, boasting cell-type diversity and a tendency toward vascularization [20,23,28,32,38,46,47,52,57,63,67,70,71,72].

COs exhibit functional tissue organization with distinct cell layers [43], spontaneous beating with a capacity for master rhythm synchronization, regeneration while in fetal phenotypes, and maturation into adult phenotypes via fatty acid oxidation (FAO) and even EHT-assisted physical maturation regimes [24,36,45,74] (Figure 5). COs recapitulate both cardiogenesis and CHD pathogenesis for modeling and drug screening [7,34,36,45,46]. COs demonstrate striking angiogenic potential in vitro and attain perfuse vasculature in vivo, supporting organoids as a tool for vascularizing CCTs. However, full in vitro perfusion is yet to be achieved [23]. VEGF-A and PDGF-BB supplementation following cardiac mesodermal induction produces EC and VSMC organoids, respectively [36,37,45,50]. PDGF-BB induces angiogenesis and vascular maturation in/around heart implants [15], even improving the induction [60], contractile force, and ECM synthesis of VSMCs in COs [20]. Given EC induction with VEGF-A in COs [41,56,72,86], supplementation with PDGF-BB and other underexplored factors has the potential to induce fluid-perfusable vasculature through the VSMC/PC reinforcement of vessels [20] (Figure 4).

Assembloids are fused multi-organoid systems capable of capturing advanced biological organization [43,44,48] and even complex neurological diseases in brain assembloids [40]. Schmidt et al. separately grew chamber-specific atrial, ventricular, and nodal organoids, then co-cultured them into a fused nodal–atrial–ventricular assembloid containing the three substituent organoids with a shared lumen/chamber [56,60]. Dedicated vascular, immune, and neural fusion organoids (FOs) can be added to COs and CCTs to achieve more diverse tissues [24,39,40,41,43,48,87,88]. FOs are often differentiated into a dedicated cell type, for instance, ECs and/or VSMCs in vascular FOs [24,37,45,50,87], MF progenitors in immune FOs [48,49], and neurons in neural FOs [40,88]. FOs enable precise tissue addition for improved complexity, specificity, and function, even in non-assembloid CCTs [24,36,43,87]. Using vascular FOs composed of iPSC-ECs and/or -VSMCs as surgical transplants is an idea with growing popularity [7,8,23,30,41,48,65]. Assembloids broadly offer complex spatial organization; drug-screening capabilities [36,40]; and the integration of vascular, immune, and nerve tissue [36,39,61,88] and thus hold promise in improving iPSC-CCT platforms.

### 2.3. Heart-on-a-Chip (HoC)

Organs on a microfluidic chip, specifically, HoCs, are an emerging technology leveraging microfluidic precision for cardiac tissue research. HoCs provide multifaceted cues essential to capturing native tissue development in vitro [4,21,24,25,37,39,43,46,66]. HoCs offer fine-tunable substrate chemistry, stiffness, and precise architecture achieved via CAD modeling and 3D printing [4,24,25,28,37], obtaining structural resolution down to around a 10–20 um channel diameter [37,80]. In-built HoC electrodes not only offer electric tissue stimulation but are very beneficial for data acquisition as sensors [4,21,28,45], enabling real-time data collection [21,34,46]. HoCs enable the precision control of chemical and gas gradients, capable of inducing solute-directed spatial patterning in tissues [3,4,21,24,46]. Combining these advances and coupling separate organoids together are organ-system-on-a-chip or body-on-a-chip (BoC) technologies. BoCs connect several organ tissues through fluidic and solute exchange [3,4,24,39,43,45,61,75], offering the particularly exciting possibility of pharmacological modeling and organ–organ crosstalk [4,21,28,45]. As a leading candidate for body-wide in vitro models [14,24,39,45], researchers are particularly interested in heart, liver, and kidney BoCs to model the effects, metabolism, and systemic cycling of drugs. Recapitulating such major vascular hubs offers an avenue for productive future research [24,34,39,45,70] (Figure 3). While cardio-pulmonary organoids [43] and cardio-hepatic BoCs have been developed, renal tissue integration is an area of ongoing research [24,39,87].

HoCs are capable of capturing several features of physiological tissue, but contributions are particularly innovative in studying and producing vascularized CCTs [3,24,30,34,39,46,58,75,76]. Fluid driving with peristaltic, acoustic, and pressure-driven pumps offers the direct emulation of vascular perfusion. Modeling vascular perfusion promises to provide key insights into vascular development, perhaps even quantitatively, especially for flow-mechanosensitive ECs and associated Notch signaling [3,34,45,61,76]. Obtaining vascular perfusion in vitro remains challenging, but HoCs can replicate perfusion by directing flow across specific cell populations [24,30,34,37,45,46,58,61,65,75,76] (Figure 4), which can also extend the lifetime of in vitro tissues [43,46]. Producing gas gradients in HoCs may offer insight into therapeutic mechanisms. Several gasses are studied in heart medicine, and HoCs offer a platform for their integration with in vitro tissues. Nitric oxide/NO is a cardioprotective, vasodilatory gas implicated in angiogenesis and released during vascular perfusion in ECs [65]. Reduced nitric oxide synthase levels are detected in CCTs after damaging inflammatory stimuli [65] and nitric oxide-releasing hydrogels show promise in treating MI [11]. Interestingly, common toxic gasses in small quantities like carbon monoxide and hydrogen sulfide can have cardioprotective effects. The release of these gasses from implanted hydrogels can combat inflammation and apoptosis in MI [11]. Perhaps such gasses could be directly introduced into cultures to simulate mechanotransduction and/or cardioprotective cues, for instance, in modeling vascular perfusion via fluid flow and nitric oxide exposure.

## 3. Applications

### 3.1. Organ Modeling

iPSC technology has significantly advanced organ modeling [17,18,23,31,32,34,38,39,42,43,45,46,47,50,51,52,54,55,56] (Figure 3). With pluripotency equal to ESCs, iPSCs can potentially become any native tissue type given appropriate signals [4,39,44,75]. Many iPSC-CCT protocols use germ layer induction followed by organ specification [8,9,31,39]. Three germ layers compose the native developing gastrula. The ectoderm, which becomes the brain and skin, the mesoderm, which becomes the heart (the first functional organ to emerge [9]), and the endoderm, which becomes many of the trunk organs [31,39,42,52,72,89]. Paralleling embryogenesis has improved the study of heart field specification and anatomy. The developing heart emerges as multiple heart fields. The first heart field (FHF) forms the heart tube [9,47,55,56,60,73] and becomes the left ventricle and part of the atria [9,28,31,63,67,70,71,86]. The second heart field (SHF) forms the right ventricle, atrial components, and the outflow tract (OFT) [9,45,47,55,60,73] (Figure 1). Nodal cell progenitors contribute to the neural crest and the juxtacardiac field (JCF), which gives rise to the epicardium [9] and contributes to the atrioventricular canal (AVC) [9,63]. iPSC-derived CCT (iPSC-CCT) differentiation works similarly, initiating mesodermal commitment followed by cardiac specification. This produces life-like models that are useful in studying cardiogenesis [36,51,55,70]. Recapitulating embryonic-like development provides a detailed in vitro model of cardiogenesis that can be manipulated to study cardiac physiology, as well as disease pathology [7,8,28,30,34,39,42,51,52,56,60,70,71,72,73,90].

iPSC-CCT and CRISPR technology have revolutionarily contributed to disease modeling [4,5,7,34,36,39,40,46,47,67]. MI-induced ischemia–reperfusion (IR) injuries and associated maladaptive fibrotic remodeling have been studied in great detail using iPSC-CCTs [4,5,6,7,16,28,34,36,39,41,45,46,56]. MI models frequently use hypoxia chambers or pouches [4,5,16,38,83]; hypoxic noradrenaline/norepinephrine; or other hormonal treatments [7,16,34], cryoinjuries [7,46,56], and/or TGF-β treatments to induce fibrosis [48]. Other coronary artery diseases have been explored with iPSC-CCT models such as atherosclerosis, stenosis, and hypertension, which increasingly emphasize VSMC inclusion alongside CMs and ECs to capture vascular pathology [37,45]. iPSC-CCTs have been extensively used to study myopathies [4,5,14,34,39,41,46,67], particularly hypertrophic (HCM) and dilated cardiomyopathy (DCM) [3,4,5,14,34,37,41,46,53,67,79,80,81,91]; channelopathies like long- (LQTS) and short-QT syndrome (SQTS) [3,4,5,30,34,37,41,45,46,53,67,80]; and even metabolic disorders like diabetes and/or mitochondrial, lysosomal, and glycogen storage disorders [4,5,8,23,34,37,39,47,67]. Following the COVID-19 pandemic, research on infection, inflammation, and cytokine storms has gained popularity, contributing to immunized CCTs and signaling studies [4,6,18,20,22,28,39,41,44,48,57]. These more comprehensive models improve disease understanding and aid drug development.

### 3.2. Drug Discovery

The evolution of disease models via iPSC-CCTs and gene editing has given rise to innovative drug discovery platforms [4,5,7,28,31,34,36,44,46,56] (Figure 3). Conventional approaches like 2D tissue culture and animal testing often fail to accurately capture drug effects [3,4,5,7,16,24,28,36,45,46,47,49,66,87], contributing to the high [4,28,46,67,86] 80–90% attrition rate of drugs through clinical trials [17,45]. Animal models have disanalogous biology (cardiac myofilaments and electrophysiology in rodents differ dramatically from humans [34,76]), raise species concerns, and often provide non-translatable outcomes [3,9,16,21,34,36,45]. This is especially so for cardiotoxicity in chronically and/or systemically administered drugs [4,8,16,34,36,41,45,46,54,67,68]. To this end, iPSC-CCTs achieve increased drug sensitivity/receptivity given their 3D nature, improving the detection of cardiotoxic compounds in screenings [17,44]. Complicating the matter, the off-target effects of an atrial drug may greatly harm ventricular tissue and vice versa [74]. To ensure cross-chamber compatibility, factor supplementation can be used to induce mixed, ventricular (via Notch signaling), or atrial (via retinoic acid supplementation) CM phenotypes [25,59,60]. A litmus test for a model’s detection of cardiotoxicity may involve screening for hypoxia-induced doxorubicin (Dox) toxicity [14,16,34,54], other conditionally cardiotoxic chemotherapeutic drugs [1,21,41,69] like sunitinib [1], or even catecholamine toxicity [9].

As a promising alternative, iPSC-CCT-based platforms enable more effective and ethically sound (3R principles: reduce, refine, and replace animal experimentation [34]) drug discovery by recapitulating life-like human biology, thereby minimizing late-stage surprises in clinical trials [3,4,5,21,43,56,68,90]. iPSC-CCTs offer several improvements to existing drug discovery schemes. Using factor-supplemented media, CCTs like CSs and COs can be produced on a batch-scale in bioreactors or well systems for high-throughput drug screening (HTDS) [1,10,17,18,34,43,44,56,65,85,90] (Figure 3). Self-assembly parallels the in vivo co-emergence of cell types, thus benefiting from being more life-like than monoculture or even multilineage co-cultures, thereby offering key insights into organogenesis and regeneration [7,34,43,46,65]. Reprogramming primary cells into iPSCs enables HTDS to be personalized on a patient-by-patient basis for the most effective treatment [2,3,4,5,21,34,36,44,45,50].

A powerful example of iPSC-CCTs in drug discovery comes from a series of Mills et al. publications in 2017, 2019, and 2021 using COs for HTDS. Their platform uses iPSC-derived COs, the coupling of tissue to physical stretchers, and the supplementation of culture media with BSA-FAs for metabolic maturation. This multifaceted platform reflects field-wide advancements, using biological, chemical, and physical methods to produce life-like tissues. After the platform was developed in 2017, researchers screened >100 cardio-regenerative compounds in 2019, successfully identifying drugs that induce the proliferation of otherwise quiescent CMs. Here, drug screening also contributed to our mechanistic knowledge, implicating the mevalonate signaling pathway in CM proliferation. Demonstrating adaptability and reproducibility, Mills et al. applied their established CO-HTDS platform to the COVID-19 pandemic, screening promising therapeutic compounds through 2021 from FDA drug libraries. Bromodomain and extraterminal family inhibitors (BETis) were discovered to rescue cardiac function from inflammation-induced dysfunction, specifically following infectious cytokine storms. Such screenings advance our understanding of pathogenesis, especially the role of inflammation in CVD. This platform synthesizes recent iPSC-CCT innovations to produce life-like tissue at a high enough throughput and scale to accommodate HTDS. Even in its infancy, CO-HTDS is capable of screening potentially hundreds of compounds. As HTDS is further developed, we can expect iPSC-CCTs to play an increasingly influential and productive role in drug discovery [17,18,44].

### 3.3. Regenerative Medicine

Cells, factors, and stimuli that induce specific tissue growth and development in CCTs could be applied as therapies to regenerate that same tissue in patients. iPSC technology offers such tools for manipulating cell fate, which are useful for regenerative medicine [76]. Cardiac regenerative medicine seeks to replenish poorly proliferative CMs, regenerate supporting tissue, and re-establish vasculature to improve heart function [76]. Pro-regeneration signals can be delivered through iPSC-CCTs via cellular and acellular constructs using various degrees of scaffold usage versus self-assembly [7,8,15,35,36] (Figure 3). Hydrogel CCTs frequently use controlled degradation with loaded cells and/or factors, sometimes spatially organized, to facilitate the recovery of damaged tissue [2,11,15]. Several ongoing clinical trials deliver iPSC-CMs or iPSC-CCTs via injection, intravenous administration, and transplantation [1,8]. Even so, conventional cell therapies struggle in clinical translation because of rapid cell death upon transplantation [10,30,34,35].

Transplanting tissues with diverse cell types, especially self-assembled CCTs like COs/organoids, has promise in overcoming low cell viability and enhancing host integration compared with cell solutions and hydrogels [1,7,30,36,38,48,83]. Cell-seeded hydrogels struggle to obtain physiological cell density, but these cell–cell interactions are recapitulated when iPSCs are guided to assemble into cohesive tissues with multiple cell types. The result is the improved viability of transplanted tissue and cellular communication with the host. Compared with monoculture transplants, cellular diversity demonstrates improved transplant viability and repair of IR injuries in several MI models [7,23,30,36,38]. Though CCTs and iPSC-CCTs in preclinical studies are primarily transplanted into immunodeficient mice, rats, and pigs, transplanting autologous iPSCs holds promise in reducing immunogenic risk in humans without systemic immunosuppression [1,7,23,30,38,39,48,67,83]. Hypoimmunogenic iPSC-CCTs have been developed using PD-L1 overexpression or MHC-II underexpression [34]. Similar to the suppression of MHC-encoding *HLA-I* and *-II* genes, *HLA-E* overexpression has also been suggested [1]. Excitingly, MHC-matched allogenic iPSC-CM injections survive for up to 12 weeks and improve MI recovery in cynomolgus monkeys. While non-fatal arrhythmia occurrence peaked 14 days after transplantation, integration with host CMs was observed, and this rate fell with time, possibly as a result of in vivo transplant maturation [27]. A key success, host vascular integration is achieved using “biologically talkative” CSs or COs, whose robust paracrine signaling owes in part to crosstalk between cell types as the tissue develops [23,38,50,61,67]. This has contributed to burgeoning clinical trials for iPSC-CCTs [1,8], including patches for treating ischemic myopathy [1,23], engineered myocardium for ventricular assistance [1], and CSs for treating HF [1,39]. These methods represent significant advancements in the practicality and efficacy of cell therapy.

Acellular alternatives have also been explored to circumvent these conventional difficulties of cellular regenerative medicine. Stem cells were once thought to seed new tissue but are now thought to function primarily through paracrine signaling via secreting extracellular vesicles, particularly exosomes. Exosomes are protein-modified lipid vesicles containing growth factors, cytokines, miRNAs, and other compounds with potent regenerative, anti-inflammatory, and pro-angiogenic effects on damaged tissue [3,8,33,35,91]. Exosomes secreted by explanted cardiac stem cells have been used [33,91,92], but recently, exosome therapy (ExT) derived from iPSC cardiac tissue exosomes has demonstrated therapeutic promise in studies and ongoing clinical trials [3,35]. Conditioning cardiac stem cells and iPSC-CCTs with culture conditions can even produce exosomes tailored to heart healing [4,33,35,91,92]. ExT boasts less invasive administration compared with surgical implantation, instead offering the intravenous and catheter-guided injection of heart- and injury-homing exosomes [3,10,33,35,78]. ExT circumvents many challenges of cell therapy like viability, immunogenicity, and transplant arrhythmia or teratoma [10] while also retaining similar comprehensive benefits [8,35]. These cell-derived therapies have seen tentative success as many exosome-loaded cardiac patches undergo clinical trials [3], with others in development [93].

## 4. Conclusions and Perspective

Patient-specific iPSCs in tandem with CRISPR gene editing enable personalized cardiogenesis and pathogenesis models. iPSC-CCTs have revolutionized CVD study through the emergence of 3D tissue structures and diverse cell types that capture the complex physiology of the heart. CCTs emphasize physiological relevance, making them well suited to organ modeling, drug discovery, and regenerative medicine (Figure 2). EHTs are scaffolded CCTs that excel in organ modeling via electromechanical stimulation and dramatic tissue maturation [4,5,9,10,24,30,31,43,46,53,54,74,75,79,81,94]. On the other hand, COs are scaffold-free/-seeded CCTs that excel in drug discovery via biochemical self-assembly, capable of batch-scale tissue production for HTDS [1,7,8,14,23,25,28,34,36,43,44,45,46,47,56,75,87]. COs, as regenerative transplants, overcome poor cell viability in cell therapy through robust tissue crosstalk and abundant cell–cell interactions [1,7,30,38,48]. Furthermore, HoCs synthesize EHT and CO technology for intensely constructed but especially life-like heart models [4,30,45,58]. Though impressive, iPSC-CCTs have many challenges to overcome. Replicating complex macro- and microscale tissue architecture can be difficult for individually fabricated EHTs however, these structures can be directly obtained from dECM [41,65]. Most platforms either sacrifice experiment volume for model complexity or vice versa. While COs have the potential to become high-throughput platforms, improving consistency between replicates is essential for downstream applications [23]. HoCs are highly complex, requiring considerable engineering expertise and resources to fabricate [21]. iPSC-CCT vascularization, immunization, and maturation are highly beneficial areas of future research for overcoming existing challenges and facilitating clinical translation, particularly in achieving maturing vasculature to increase CCT size and complexity. Comparisons and summaries of iPSC-CCT platforms can be found in Table 1.

Producing vasculature in vitro is essential for increasing iPSC-CCT size and complexity [1,8,10,20,24,28,36,38,43,44,46,48,61,65,67,75,76]. Without perfusion, hypoxic necrosis typically occurs beyond 0.1 mm of depth in dense tissues [24,25] and 1–2 mm in COs [23]. Many COs achieve microvascular networks of ECs and partial mural populations via VEGF-A and/or bFGF supplementation [23,43,47,51,56,57,61,65,67,71,73,86,87], yet work remains for mural VSMC/PC reinforcement [11,20,32,48,61,65]. PDGF-BB and bFGF co-treatment induces the emergence of mesenchymal tissue and vascular cell types [20,89]. Reflecting this, MSCs promote cardiovascular maturation and angiogenesis [3,11,12,26,46,48,58,65,78,80,93,98,99]. PDGF-BB is underexplored but promising for mural induction [10,11,15,20,32,98], as evidenced by its potent mitogenicity [15,98], therapeutic heart revascularization [11,15,20], and VSMC induction in organoids [20,50,60] (Figure 4).

Immunization is also critical for iPSC-CCT models, as regulating inflammation and immunity is a key goal in heart medicine [4,6,7,18,19,24,34,36,41,44,48,61,65,100,101]. Understanding inflammation is essential to studying CVD but cannot be accurately recapitulated in vitro without immunization given the heart’s resident MF population [41]. Inflammation-inducing factors include TNF/TNF-α, IFN-γ, IL-1β, IL-6, poly(I:C), and LPS [4,5,11,12,13,18,19,20,22,41,45,49,77,84,100,101]. Co-culture with MFs is attractive because they regulate inflammation, promote angiogenesis, aid cardiac tissue maturation (including via IGF-1 release), and facilitate CM electrical integration via Cx43 induction [4,19,38,41,45,48,65,100] (Figure 5). To this end, iPSC-derived monocytes can provide a reliable source of MFs for immune co-culturing [41,45,49,77,84,85].

Mature CMs are non-proliferative. To obtain cardiac tissue in vitro without explant, “maximally immature” pluripotent starting populations are used. Thus, iPSC-CMs initially have a fetal phenotype, often lasting several weeks, challenging adult heart disease models and transplants [2,7,8,10,16,17,24,26,30,31,34,36,46,53,58,79,81,82,94]. Transplanting immature iPSC-CCTs carries arrhythmia risk from improper CM integration [10,24,30,31,36,46]. Adult CMs switch from glucose to FAO for energy, grow/fuse mitochondria, align sarcomeres, and elongates the cell. Several methods induce CM maturation [5,10,30,31,36,81,94], including cultures with multiple cell types (e.g., ECs, CFs, VSMCs/PCs, MSCs, and MFs) [2,4,10,14,30,36,45,54,79,80], BSA-bound fatty acid supplementation (BSA-FAs) (commonly, palmitate- and oleate-BSA [20,31,79]) for metabolic maturity with galactose for non-CM survival [4,10,17,18,20,36,45,53,54,79], thyroid hormone triiodothyronine (T3), glucocorticoid dexamethasone (Dex), IGF-1 [5,8,10,30,31,36,46,58,80,82,94], mechanical stretching (commonly 10–20% elongation, 1–3 Hz), and electrical pulses (commonly 3–5 V/cm, 2–6 Hz) [5,9,10,30,31,54,74,81,94]. These methods achieve increased contraction force and positive force–frequency relationships. As CMs mature, T-tubule network formation and calcium-handling protein expression enable excitation–contraction coupling, which reduces spontaneous contraction [4,5,10,17,20,24,30,31,36,41,45,46,53,58,67,74,79,81,82,94] (Figure 5). For disease models and drug discovery, an adult tissue phenotype is essential given the presence/absence of proliferation in fetal/adult CMs, respectively, especially so for studying heart regeneration [31]. Electrophysiology is another essential consideration, as mature CMs express dramatically different ion channel profiles than immature CMs. Prominent ion channels in mature CMs include those for potassium (maintains resting potential) KCNJ2 and KCNJ12, sodium (initiates depolarization) like SCN5A, and calcium (extends AP duration) RYR2 and SERCA2 [31]. In regenerative transplants, CM maturity is essential for host integration and avoiding complications like arrhythmia. Hallmarks of host integration include CM–CM and CM–CF electrical coupling with Cx43 [29,44,54] and CM–CM mechanical coupling with intercalated discs (ICDs) [31].

Taking a comprehensive look at iPSC-CCTs, we find powerful platforms for various applications. Despite existing outside of whole-body systems like animal models, iPSC-CCTs provide an indispensable view of human biology. These platforms allow researchers to investigate cardiac physiology, pathology, and CVD therapies before clinical trials. COs defy the poor cell viability of cellular therapies, featuring multi-cell-type crosstalk which improves tissue maturity [7,30,43,76,79], biocompatibility, and in vivo host integration [1,7,30,38,48,83]. Transplanted COs attenuate cardiac damage and integrate with host vasculature in immunodeficient mice MI models, a major step in viable cell therapies [23,30,38,50,61,67] Cell-derived ExT is acellular and negligibly immunogenic compared with cell therapy. iPSC-CCTs and cardiac tissue stimulation produce exosomes tailored to specific tissue healing [3,33,35,91,92,93,99]. Similar to ECM-targeting peptides used for wound healing, like PLGF-2 derivatives [98,102], cardiac homing peptides (CHPs) specifically target ischemic myocardial tissue in ExT, improving MI recovery [33,35,78,92,93,103]. Non-invasive delivery with catheters, intravenous injections, and nebulizers facilitates the clinical translation of ExT [93,99]. While many iPSC-CCT platforms are early in development, their undeniable potential makes them enticing tools for the future of cardiovascular research.

## Figures and Tables

**Figure 1 cells-13-00250-f001:**
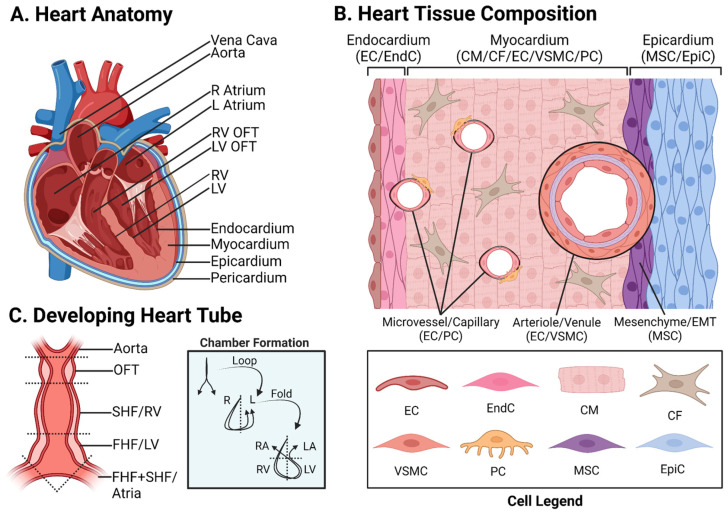
Heart Composition: (**A**) The adult human heart is divided via the septum into left and right halves. Each side has an atrium, ventricle, and OFT, with the muscular left ventricle pumping blood through the aorta out to the body. The heart has three layers with an external sac, the pericardium. (**B**) Adult human heart tissue contains three layers: the endocardium, composed of ECs and EndCs; the myocardium, composed of CMs and CFs with vessels composed of ECs, VSMCs, and PCs (microvessels and capillaries have ECs and PCs; larger macro-vessels like arterioles and venules have ECs and VSMCs); and the epicardium, composed of MSCs and EpiCs. The EMT produces mesenchymal tissue below the EpiCs. (**C**). The embryonic heart emerges from the gastrula mesoderm as a fused vessel known as the heart tube, which contains the SHF, producing the RV, and the FHF, producing the LV. As cardiogenesis progresses, the heart tube loops, folds, and develops a septum, eventually producing the four heart chambers. CF, cardiac fibroblast; CM, cardiomyocyte; PC, pericyte; MSC, mesenchymal stem cell; EC, endothelial cell; EndC, endocardial cell; EMT, epithelial-to-mesenchymal transition; EpiC, epicardial cell; FHF, first heart field; OFT, outflow tract; L, left; LA, left atrium; LV, left ventricle; RA, right atrium; RV, right ventricle; R, right; SHF, second heart field; VSMC, vascular smooth muscle cell.

**Figure 2 cells-13-00250-f002:**
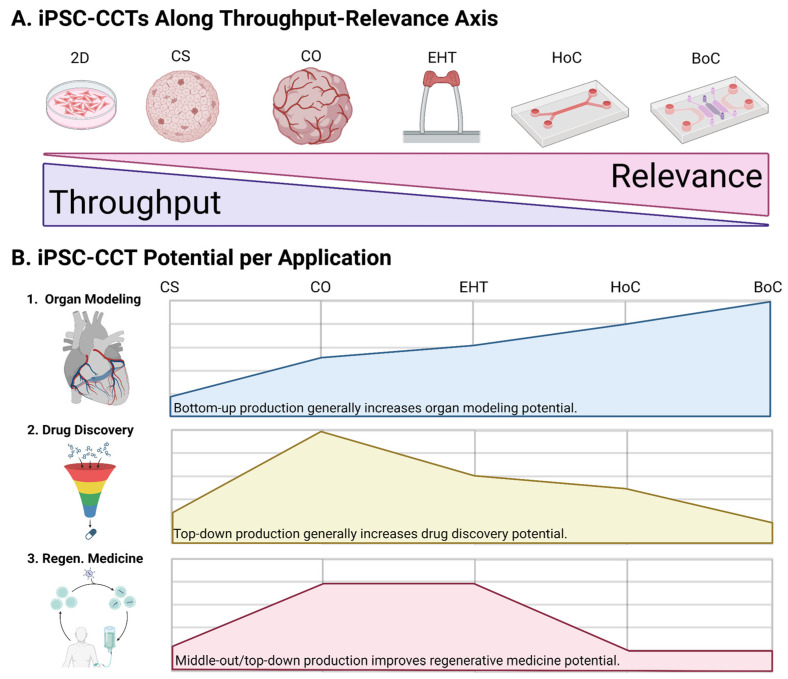
Constructed Cardiac Tissues: (**A**). iPSC-CCTs are 3D tissues that produce more physiologically relevant, life-like cardiac tissue than 2D tissue cultures, which can be used to study physiology, test therapies, and be developed for transplantation. These emerging platforms can broadly be arranged by emphasis on high production volumes, that is, high-throughput, or on high physiological relevance. (**B**). Top-down strategies enable batch-scales, often leveraging iPSC self-assembly to produce tissues with diverse cell types using minimal intervention during culture. This is particularly useful for drug discovery and personalized screenings. Bottom-up strategies involve the modular assembly of cells and components, usually into scaffolded tissue, which enable the finetuning of cardiac tissues. This is particularly useful in organ modeling, including modeling healthy versus diseased tissue. Combining middle-out production methods and top-down self-assembly is promising for regenerative medicine, which benefits from biological complexity but also scalable production. 2D, two-dimensional; 3D, three-dimensional; BoC, body-on-a-chip; CCT, constructed cardiac tissue; CO, cardiac organoid; CS, cardiac spheroid; EHT, engineered heart tissue; HoC, heart-on-a-chip; iPSC-CCT, iPSC-derived CCT; iPSC, induced pluripotent stem cell.

**Figure 3 cells-13-00250-f003:**
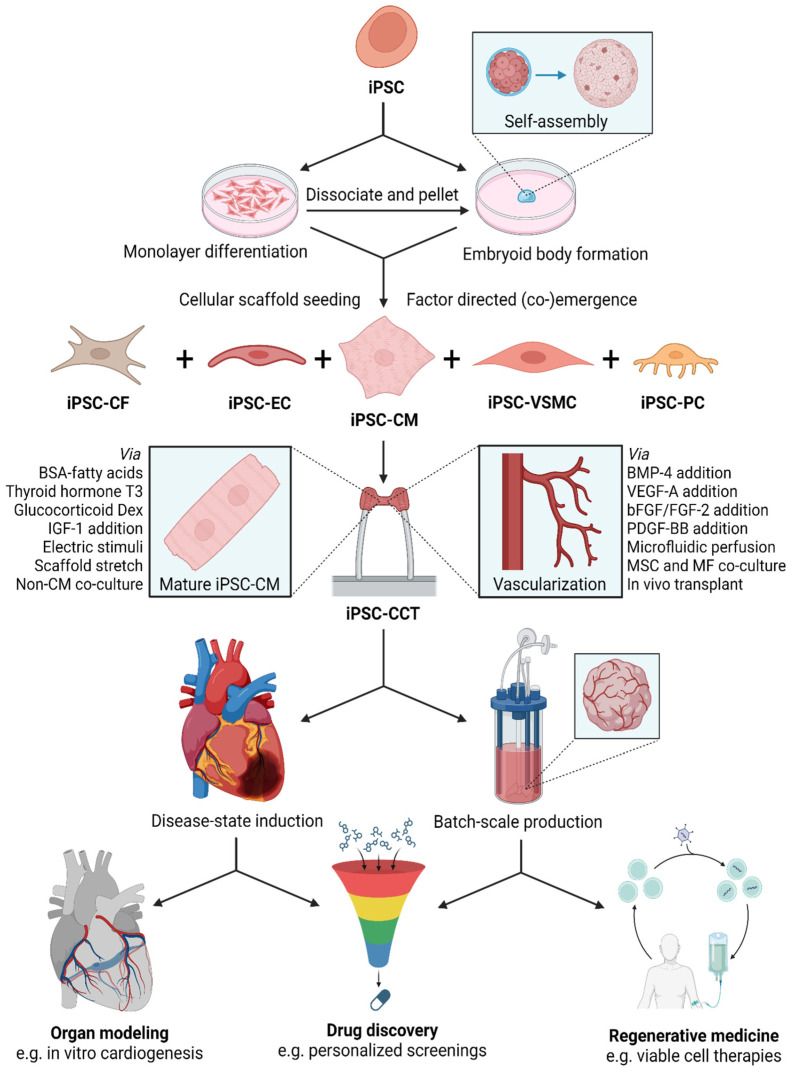
iPSC-CCT Application Development: iPSC-CCTs can be developed for use in organ modeling (including disease modeling), drug discovery, and regenerative medicine, offering unique advantages in each application. For instance, developmental cardiogenesis models of healthy versus diseased tissue for organ modeling, patient-specific screenings for drug discovery, and crosstalk-rich tissue transplants with good biocompatibility and promising host integration for regenerative medicine. iPSCs, either from an established or specific patient cell line, are either pre-differentiated as a monolayer before dissociation and scaffold seeding, or pelleted into a self-assembling EB to begin production. Certain cellular scaffolds like EHTs provide good iPSC-CM maturation, and factor-directed co-emergence, as used in COs, can induce tissue vascularization. Certain production and induction techniques can be used to develop cardiac tissues whose physiology may, for instance, be directly studied, used to screen large numbers of candidate drugs, and assembled into regenerative transplants with better-than-usual cell viability. bFGF, basic fibroblast growth factor, synonymous with FGF-2; BMP-4, bone morphogenic protein four; BSA, bovine serum albumin; iPSC, induced pluripotent stem cell; CCT, constructed cardiac tissue; CO, cardiac organoid; Dex, dexamethasone; EHT, engineered heart tissue; EB, embryoid body; iPSC-VSMC, iPSC-derived vascular smooth muscle; iPSC-CCT, iPSC-derived CCT; iPSC-CF, iPSC-derived cardiac fibroblast; iPSC-CM, iPSC-derived cardiomyocyte; iPSC-EC, iPSC-derived endothelial cell; iPSC-PC iPSC-derived pericyte; IGF-1, insulin-like growth factor one; MSC, mesenchymal stem cell; MF, macrophage; PDGF-BB, platelet-derived growth factor B form dimer; T3, triiodothyronine; VEGF-A, vascular endothelial growth factor form A.

**Figure 4 cells-13-00250-f004:**
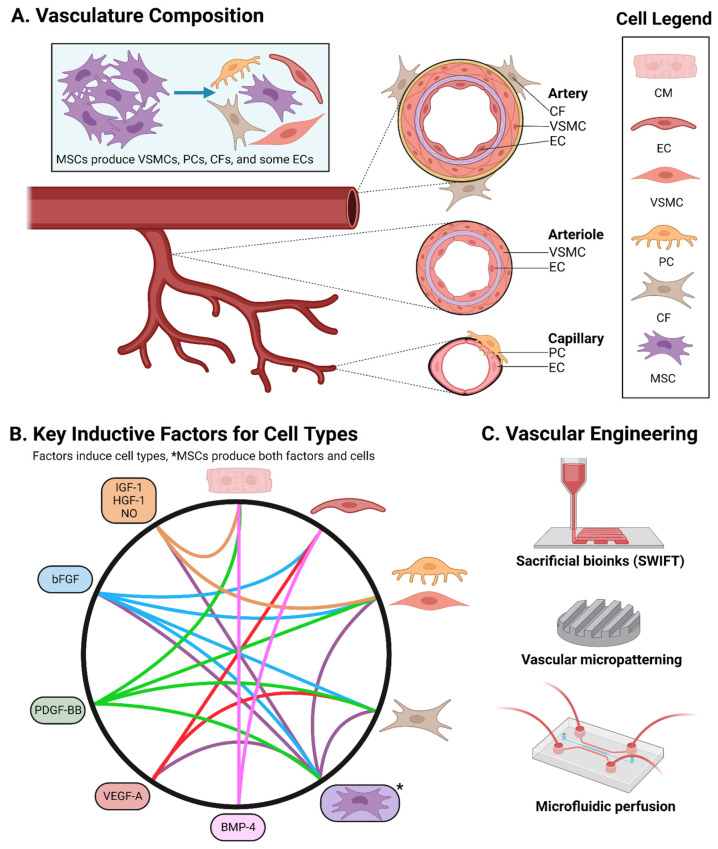
Vascularization Methods: (**A**). Vascularizing iPSC-CCTs is essential to improving their size and complexity through increased oxygen and nutrient availability. Native vessels come in various sizes and structures: arteries are large and possess external connective tissue layers; arterioles are small and also have an endothelium reinforced with smooth muscle; and capillaries are very small, with an endothelium supported by a basement membrane and PCs. (**B**). Factor addition is useful for vascularization. Inductive factors critical to respective cell types are as follows: BMP-4 for CMs and ECs; VEGF-A for ECs; PDGF-BB for VSMCs/PCs and MSCs; and bFGF for CFs and MSCs, while IGF-1, HGF-1, and nitric oxide/NO support CMs and EC-VSMC crosstalk. (**C**). Engineering approaches to vascularization include sacrificial bioinks (SWIFT) that dissolve away to leave channels for flow, branching vascular-like micropatterning, and the direct emulation of perfusion in a microfluidic chip. bFGF, basic fibroblast growth factor, synonymous with FGF-2; BMP-4, bone morphogenic protein four; CCT, constructed cardiac tissue, CF, cardiac fibroblast; CM, cardiomyocyte; EC, endothelial cell; HGF-1, hepatocyte growth factor one; iPSC-CCT, iPSC-derived CCT; iPSC, induced pluripotent stem cell; IGF-1, insulin-like growth factor one; MSC, mesenchymal stem cell; NO, nitric oxide; PDGF-BB, platelet-derived growth factor B form dimer; PC, pericyte; SWIFT, sacrificial writing into functional tissue; VEGF-A, vascular endothelial growth factor form A; VSMC, vascular smooth muscle cell.

**Figure 5 cells-13-00250-f005:**
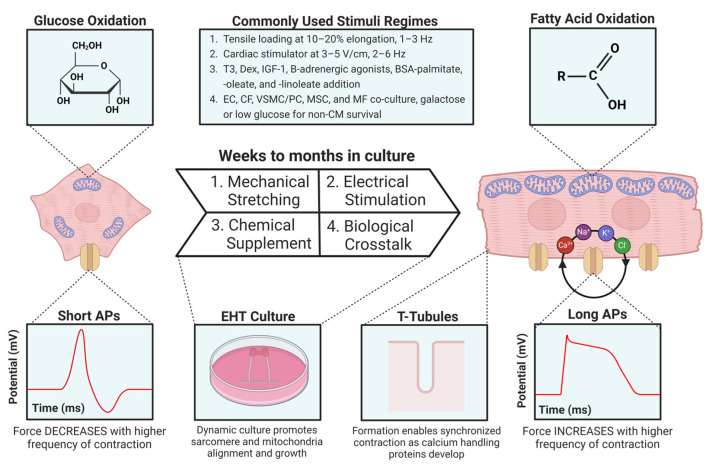
Maturation Methods: iPSC-CM maturity is a critical feature of iPSC-CCT use in adult disease models, drug discovery, and the electromechanical integration of regenerative transplants into the host myocardium. CMs lose their proliferative ability with phenotypic maturity. Mature CMs have elongated morphology, aligned sarcomeres, binucleation, physiological hypertrophy, enlarged mitochondria performing primarily FAO, T-tubule networks for synchronized contraction, and proteins for calcium handling, resulting in longer APs with a positive contractile force–frequency relationship. Maturation is induced through various techniques: 1. mechanical stretching to 110–120% tissue length (length is often ramped up over time to increase CM length) at 1–3 Hz; 2. electrical cardiac stimulation with 3–5 V/cm at 2–6 Hz (frequency is often ramped up over time); 3. chemical supplementation with T3, Dex, other glucocorticoids, IGF-1, sometimes β-adrenergic agonists like adrenaline, and BSA-bound fatty acids to facilitate a switch from glucose to FAO metabolism; and 4. co-culture with diverse cell types, which increases iPSC-CM maturity and β-adrenergic/cAMP signaling. AP, action potential; BSA, bovine serum albumin; cAMP, cyclic adenosine monophosphate; CF, cardiac fibroblast; CM, cardiomyocyte; Dex, dexamethasone; EHT, engineered heart tissue; EC, endothelial cell; FAO, fatty acid oxidation; iPSC-CM, iPSC-derived CM; iPSC, induced pluripotent stem cell; IGF-1, insulin-like growth factor one; MSC, mesenchymal stem cell; MF, macrophage; PC, pericyte; T3, triiodothyronine; VSMC, vascular smooth muscle cell.

**Table 1 cells-13-00250-t001:** Representative examples of iPSC-CCT platforms. Pros, cons, and features of platforms. Listed are notable iPSC-CCT publications and the first author, followed by that respective platform’s advantages; disadvantages; a description of the cardiac tissue architecture; vascularization methods or method of modeling vascular interactions; and maturation methods of improving the phenotypic maturity of CM electrophysiology, metabolism, and morphology.

iPSC-CCT Platform	Author	Pros	Cons	Architecture	Vascularization	Maturation
Cardiac Spheroid	Arhontoulis et al., 2022 [22]	Crosstalk between multiple cell types captures inflammatory signaling Sufficient throughput for effective drug screening	Sequential introduction of cell lines limits the emergence of structures	Aggregate of iPSC-derived cardiomyocytes, and established cardiac fibroblast, vascular stem cell, and mesenchymal stem cell lines	Inclusion of vascular stem cells, while not itself forming vasculature, nonetheless captures physiological responses to injury and infection like endothelial damage	Crosstalk between cell types upregulates adrenergic signaling, driving maturation, though culture periods were less than a week, limiting observed maturation
Cardiac Spheroid to Organoid	Tan et al., 2023 [38]	Highly biocompatible given the presence of multiple cell types Perfusable, host vascular integration upon transplant in infarcted, immunodeficient mice Nanowires improve host electrical integration	Limited structural complexity	Organized aggregate of iPSC-derived cardiomyocytes, human cardiac fibroblasts, and established vascular stem cell lines, with optional nanowires	Inclusion of vascular stem cells and continued culturing primes the spheroid/organoid to promote angiogenesis at the infarcted site of transplantation	Crosstalk between cell types upregulates adrenergic signaling, driving maturation within the spheroid/organoid and with host tissue
Cardiac Organoid	Lewis-Israeli et al., 2021 [47]	Self-assembles into all heart layers Epicardial spreading and microvascular network formation Multiple chambers per organoid Relatively inexpensive with few growth factors used	Batch-to-batch variability Difficult to apply electromechanical stimuli to batches	Microvascularized, chambered, three-layered heart miniature organ grown from iPSC aggregate	Produces microvasculature with minimal growth factors via Wnt reactivation for epicardial spreading and microvessel networks	Metabolic maturation by supplementing culture media with fatty acids to promote fatty acid oxidation in cardiomyocytes after about a week in culture
Cardiac Pulmonary Organoid	W.H. Ng et al., 2022 [95]	Captures inter=germ-layer tissue cooperation Relatively inexpensive with few growth factors used Self-directed spatial sorting of tissues from different germ lineages	Batch-to-batch variability Though iPSCs are co-differentiated, they are initially in 2D cultures, which restricts early organization	Spatially distinct heart and lung miniature organ system grown from iPSC aggregate	Angiogenic factors added early in culture but used for mesoderm and endoderm specification, thus lacking vasculature formation	Crosstalk between germ layers promotes mutual development via dual Wnt and TGF-β signaling modulation followed by retinoic acid and glucocorticoid treatment
Cardiac Foregut Organoid	Silva et al., 2021 [71]	Captures inter-germ-layer tissue cooperation Epicardial spreading and microvasculature Tissue lifespan longer than a year	Batch-to-batch variability Lack of chamber-like morphology	Beating myocardial core surrounded by non-beating epicardial layer grown from iPSC aggregate	Epicardial-permissive media with ascorbic acid after about a week in culture allows microvessels to emerge with epicardial-derived vascular cells	Crosstalk between germ layers promotes mutual development, and prolonged culture of over a year enables maturation into steady-state, chamber-specific cardiomyocytes
Cardiac Assembloid	Schmidt et al., 2023 [60]	Heart field and chamber-specific organoids Action potential propagates from atrial organoid to left and then right ventricle Flexible model; can further study individual chamber-specific organoids Low batch-to-batch variation for an organoid platform	Requires customized 3D printing of molds with silicone casting to properly align the three organoids into a linear heart-tube-like structure Absence of epicardium and microvasculature Relatively expensive with many growth factors used	Atrial, left ventricular, and right ventricular organoids use linearly inside a mold into a heart tube-like structure grown from separate iPSC aggregates	Angiogenic factor supplementation improves endocardial and endothelial emergence but without significant vasculature formation	Chamber-specific or combined media for cardiomyocyte maturation via fatty acid metabolism, hormones, and glucocorticoids
Engineered Cardiac Organoid	Mills et al., 2019, 2021 [17,18]	Self-directed formation of structure and microvessels Biologically complex for an engineered tissue Macroscale 3D structure for physiological drug response Sufficient throughput for effective drug screening	Requires specialized, expensive machinery Lack of chamber-like morphology	iPSC-derived cardiomyocytes and iPSC-derived stromal cells of epicardial lineage densely suspended in hydrogel	Inclusion of iPSC-derived vascular cells enables microvascular formation in an isogenic platform	Suspension of organoids in engineered heart tissue stretcher allows for contractile stimulation alongside eventual fatty acid supplementation
Engineered Heart Tissue	Ronaldson-Bouchard et al., 2018 [81]	Exceptional electrophysiological, metabolic, and mechanical maturation markers Mature cardiomyocytes have good drug sensitivity Ramped, non-uniform stimulation improves maturation	Requires several specialized machines Limited biological complexity with no epicardium, endocardium, or cardiac fibroblasts	Mechanically suspended cardiomyocyte and cardiac fibroblast seeded in a fibrin hydrogel	Angiogenic factors added early in culture but used for cardiac specification, thus lacking vasculature formation	Gradually increasing intensity of mechanical stretch and electrical stimulation leads to robust, anisotropic/elongated cardiomyocytes
Heart-on-a-Chip	Shin et al., 2016 [96]	In-built microelectrodes for sensing and data collection Infusion system for tested drugs Detection of molecules indicating drug cardiotoxicity Using microscale fluid volumes improves cost efficiency	Sealing and sterility in microfluidics is difficult Dedicated electronic systems required Only a single cell type dramatically limits physiological relevance	Microfluidic bioreactor culture of cardiac spheroids composed of embryonic stem cell-derived cardiomyocytes	Limited biological complexity does not enable vasculature formation	Difficulty maintaining microfluidic culture does not enable significant cardiomyocyte maturation
Heart-and-Liver-on-a-Chip	F. Yin, et al., 2021 [97]	Captures key drug interactions through the larger circulatory system Reveals drug effects as other tissues process the drug into downstream metabolites Accommodates large sample size for sensitive detection of analytes	Several custom parts required for novel co-culture system Lack of in-built sensors necessitates manual media extraction for analysis of analytes	Several iPSC-derived cardiac and hepatic miniature organs held in place with micropillars and separated by a permeable membrane in microfluidics systems	Though not possessing vascular cells, macroscopic vascular relationships are modeled	Hormonal treatment after about three weeks of culture during hepatic tissue integration and cardiac maturation

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
