# Peer review of "Advances in the Generation of Constructed Cardiac Tissue Derived from Induced Pluripotent Stem Cells for Disease Modeling and Therapeutic Discovery"

_cells, 2024, doi:10.3390/cells13030250_

Round 1

Reviewer 1 Report

Comments and Suggestions for Authors

In this review, Roland and colleagues highlight recent progress and remaining challenges in engineering 3D cardiac tissues, and its implications in disease modeling and therapeutic discovery. The review thoroughly and accurately details the benefits and shortcomings of various 3D constructed cardiac tissues (CCTs), overall presenting a strong review of the field. Although the authors address diverse concepts related to tissue design (architecture, maturity, and cellular diversity), they primarily focused on issues related to vascularization. This review can benefit from addressing the following minor comments:

Comments:

1.     The review can be strengthened by incorporating a more thorough discussion on electrophysiological challenges, including the implication of CM maturation state on ion channel differences, disease modeling/drug testing, arrhythmogenesis, and electrical integration of implanted engineered cardiac tissues. These concepts are only briefly discussed at the end of the review.

2.     Providing a detailed table summarizing the literature and pros/cons of the various CCT models in issues pertinent to vascularization/electrophysiology/architecture will be beneficial to the readers.

3.     In the abstract, the phrase “iPSC-CCTs recapitulate heart biology and integrate post-transplantation” present a bold statement that is not entirely accurate and should be revised. As detailed by the authors in the review, in vitro cardiac tissues models have yet to achieve the same level of maturity and architecture as an adult human heart. Likewise, integration post-transplantation, although can be achieved still faces many challenges, (including strong history of arrhythmogenesis/VTs in injected hiPSC-CMs).

4.     Lines 44-45 requires references.

Author Response

In this review, Roland and colleagues highlight recent progress and remaining challenges in engineering 3D cardiac tissues, and its implications in disease modeling and therapeutic discovery. The review thoroughly and accurately details the benefits and shortcomings of various 3D constructed cardiac tissues (CCTs), overall presenting a strong review of the field. Although the authors address diverse concepts related to tissue design (architecture, maturity, and cellular diversity), they primarily focused on issues related to vascularization. This review can benefit from addressing the following minor comments:

Comments:

  1. The review can be strengthened by incorporating a more thorough discussion on electrophysiological challenges, including the implication of CM maturation state on ion channel differences, disease modeling/drug testing, arrhythmogenesis, and electrical integration of implanted engineered cardiac tissues. These concepts are only briefly discussed at the end of the review.

Response: Thank you for your insight and recommendations! We have included discussion about electrical coupling through gap junctions throughout, ion channels expressed in mature CMs, and considerations of tissue maturity for modeling/testing, as well as electro-mechanical integration of CMs for transplant and arrhythmogenic considerations., lines 52-53, 119-120, 544-548, 621-625, 638-651.

  1. Providing a detailed table summarizing the literature and pros/cons of the various CCT models in issues pertinent to vascularization/electrophysiology/architecture will be beneficial to the readers.

Response: Several iPSC-CCTs between spheroids and body-systems-on-a-chip have been listed with pros/cons, as well as a description of the generated tissue structure, vascularization methods/modeling used, and maturation methods/modeling used. Table 1, lines 594-601

  1. In the abstract, the phrase “iPSC-CCTs recapitulate heart biology and integrate post-transplantation” present a bold statement that is not entirely accurate and should be revised. As detailed by the authors in the review, in vitrocardiac tissues models have yet to achieve the same level of maturity and architecture as an adult human heart. Likewise, integration post-transplantation, although can be achieved still faces many challenges, (including strong history of arrhythmogenesis/VTs in injected hiPSC-CMs).

Response: We have removed transplant claims from the abstract, as well as rephrasing similar statements in the text body for clarity. Lines 13-15, 166, 659

  1. Lines 44-45 requires references.

  Response: References added. Lines 47-49.

Reviewer 2 Report

Comments and Suggestions for Authors

In this manuscript, the authors have provided a comprehensive summary of the recent advancements in constructed cardiac tissues derived from iPSCs. The manuscript is skillfully written, addressing an important topic that holds great significance for researchers in this field. However, I would like to raise a concern regarding the comparison between iPSC-CCTs and other model systems, particularly highlighting the limitations of iPSC-CCTs.

Author Response

In this manuscript, the authors have provided a comprehensive summary of the recent advancements in constructed cardiac tissues derived from iPSCs. The manuscript is skillfully written, addressing an important topic that holds great significance for researchers in this field. However, I would like to raise a concern regarding the comparison between iPSC-CCTs and other model systems, particularly highlighting the limitations of iPSC-CCTs.

Response: Thank you for your constructive review! I have included more explicit mentions of key shortcomings in engineered heart tissues, cardiac organoids, and heart-on-a-chip, as well as highlighting the key benefit of human tissue in iPSC-CCTs in the conclusion as a point in the Conclusions and perspectives. Lines 585-595, 652-656, Table 1, and 666-669.

Reviewer 3 Report

Comments and Suggestions for Authors

Overall, the review is well structured, summarizes the current state of cardiac modeling techniques in detail, and no important details were left out of the manuscript. I have summarized my observations:

• In Figure 2, there is no mention of disease modeling as a possible use, which could at least be mentioned in the figure description (especially since it is also included in the title).

• The description of the cell types of the heart in the introduction and the formation of the heart at the end of the manuscript are more detailed than the other parts. It would be more proportionate to the entire article if I could shorten it a bit and mention the markers of the cell types instead.

• In the EHT section, he discusses in detail the various synthetic materials and mentions that they also use dECM for this purpose, but it is not clear what kind of source scaffolds are used and that this also has a beneficial effect on the maturation of CMs (for example: https://www.ncbi.nlm.nih.gov/pmc/articles/PMC4991595/). When describing bioprinting, the negatives should be mentioned, e.g. the fragility of tissue when creating larger, macroscopic models.

•It is not clear why EHT points to this text box. Perhaps starting from the text box, it should be pointed to the EHT, if the authors' idea is that all these processes can play a role during the formation of the EHT.

Author Response

Overall, the review is well structured, summarizes the current state of cardiac modeling techniques in detail, and no important details were left out of the manuscript. I have summarized my observations:

  • In Figure 2, there is no mention of disease modeling as a possible use, which could at least be mentioned in the figure description (especially since it is also included in the title).

Response: Thank you for your constructive recommendations! Disease modeling mentioned in figure label, highlighted organ modeling healthy versus diseased tissue to illustrate point. In this review, we were focused on methodological advances in iPSC-CCTs. We changed the title as well. Figure 2 and 3 (figure descriptions), lines 90-91, 97-98, 156-158, 163-165.

  • The description of the cell types of the heart in the introduction and the formation of the heart at the end of the manuscript are more detailed than the other parts. It would be more proportionate to the entire article if I could shorten it a bit and mention the markers of the cell types instead.

Response: Various less-important pieces of information removed from cell type and heart development paragraphs. Lines 103-121, 440-445

  • In the EHT section, he discusses in detail the various synthetic materials and mentions that they also use dECM for this purpose, but it is not clear what kind of source scaffolds are used and that this also has a beneficial effect on the maturation of CMs (for example: https://www.ncbi.nlm.nih.gov/pmc/articles/PMC4991595/). When describing bioprinting, the negatives should be mentioned, e.g. the fragility of tissue when creating larger, macroscopic models.

Response: Included mention of allogenic and xenogenic dECM sources and the beneficial impact of native ECM structure on CM maturation as a critical tissue engineering objective. The fragile structure and subpar overall output of bioprinter ventricles has been included. Lines 196-199, 247-251

  • It is not clear why EHT points to this text box. Perhaps starting from the text box, it should be pointed to the EHT, if the authors' idea is that all these processes can play a role during the formation of the EHT.

Response: Direction of pointing has been reversed to convey that steps can be done in the context of EHT culture. Figure 5.

Round 2

Reviewer 1 Report

Comments and Suggestions for Authors

The authors have addressed all reviewers' concerns, strengthening the review. 

Reviewer 2 Report

Comments and Suggestions for Authors

I have no other questions.

Reviewer 3 Report

Comments and Suggestions for Authors

In my opinion, the review process made the manuscript more high-quality and suitable for publication.